# Expression and Possible Role of Nicotinic Acetylcholine Receptor ε Subunit (AChRe) in Mouse Sperm

**DOI:** 10.3390/biology10010046

**Published:** 2021-01-11

**Authors:** Yusei Makino, Yuuki Hiradate, Kohei Umezu, Kenshiro Hara, Kentaro Tanemura

**Affiliations:** Laboratory of Animal Reproduction and Development, Graduate School of Agricultural Science, Tohoku University, Sendai, Miyagi 980-0845, Japan; yuki.hiradate.d4@tohoku.ac.jp (Y.H.); kohei.umezu.d6@tohoku.ac.jp (K.U.); kenshiro.hara.b6@tohoku.ac.jp (K.H.)

**Keywords:** mouse, sperm, testis, nicotinic acetylcholine receptor, acetylcholine, nicotine, sperm acrosome reaction

## Abstract

**Simple Summary:**

Numerous neurotransmitters have been reported to affect mammalian sperm function and are thought to play an important role in the phenomenon of fertilization. This study focused on acetylcholine (ACh) and its receptor subunit, the ε subunit (AChRe), and analyzed the effect of ACh and AChRe on mammalian sperm function and fertilization. The present study revealed the localization of AChRe in the murine testes and spermatozoa, and showed that acrosome reaction (AR), an important change in sperm function associated with fertilization, is suppressed by ACh. Furthermore, this suppressing effect was significantly inhibited by an AChRe specific antagonist, suggesting that AChRe play as a regulator of mammalian sperm AR. These results may aid in further elucidating the phenomenon of mammalian sperm function for fertilization by ACh and in the establishment of a new method of superior sperm selection via AChRe.

**Abstract:**

The nicotinic acetylcholine receptor (nAChR) is one of the receptors of acetylcholine (ACh), and nicotine (NIC) acts as an agonist of this receptor. Among the nAChR subunits, we found that the ε subunit (AChRe) had approximately 10 to 1000 times higher level of mRNA expression in mouse testes than the other subunits. In this study, we aimed to elucidate the expression and localization of AChRe in the testes and spermatozoa of mice and clarify the effect of AChRe on sperm function. Immunocytochemistry showed that AChRe was expressed in the murine testes and spermatozoa. We found that AChRe was localized only in elongated spermatids from step 12 onwards in the testes. In spermatozoa, AChRe was localized in the head, especially in the anterior region of the acrosome, but only approximately 50% of spermatozoa showed this immunoreactivity. Additionally, we analyzed the effects of ACh and NIC on sperm acrosome reaction (AR) and found that both ACh and NIC suppressed the AR rate, which was restored by an AChRe-specific antagonist. These results suggest that AChRe may be a regulator of mammalian sperm AR.

## 1. Introduction

Mammalian spermatozoa are produced in the testes, passed through the epididymis, and ejaculated into the female reproductive tract. Immediately after being ejaculated, spermatozoa cannot fertilize an oocyte. During migration from the vagina or uterus to the ampulla of oviduct, spermatozoa are exposed to various factors such as hormones, signal transducing molecules, enzymes, ions, and lipids secreted from the female tissues [1]. Sperm interact with these factors and undergo several biochemical changes to gain fertilizing ability (known as capacitation) [2,3,4]. Factors secreted from the female reproductive tissues promote sperm protein tyrosine phosphorylation or remove cholesterol from the sperm cell membranes [5,6], suggesting that numerous factors can modulate sperm functions, such as capacitation and acrosome reaction (AR). Recently, several neurotransmitters have been reported to be involved in these sperm functions [7,8,9,10,11].

Acetylcholine (ACh) is one of the oldest and most well-known neurotransmitters. Although ACh is commonly known as a parasympathetic neurotransmitter, it is associated with sympathetic and motor nerves. Furthermore, synthesis and release of ACh are found in various non-neuronal cells and tissues in mammals and it is collectively considered a non-neuronal ACh. Several studies have reported that non-neuronal ACh is responsible for local intercellular signaling in immune system cells, vascular endothelial cells, placenta, keratinocytes, airway epithelial cells, gastrointestinal epithelial cells, and bladder epithelial cells [12].

There are two types of ACh receptors: nicotinic (nAChR) and muscarinic (mAChR). nAChR is a pentamer consisting of α, β, γ, δ, and ε protein subunits, whereas mAChR has five subtypes (M1–M5). nAChRs are ligand-gated ion channels, and nicotine (NIC) acts as one of its agonists. They are mainly expressed in the central and peripheral nervous systems, muscles, and are involved in fast synaptic transmission [13,14]. In vertebrates, 10α, 4β, and single δ, γ, and ε nAChR subunits have been identified. Muscle-type nAChRs conform to a strict stoichiometry of either (α1)_2_ β1 γ δ (expressed in fetal tissue) or (α1)_2_ β1 γ ε (adult tissue), while neuronal-type nAChRs form a hetero-pentamer composed of α and β or a homo-pentamer composed of single α subunits. Interestingly, the subunit association fluctuates, generating a high level of functional diversity in terms of pharmacological specificities and channel permeability and kinetics. However, the role of each subunit and the interaction mechanisms between different subunits is still unclear.

Previous studies have reported that nAChRs are expressed on some mammalian spermatozoa [15,16,17], and on human spermatozoa, α3, α5, α7, α9, and β4 subunits of nAChR are localized to the midpiece, neck, and post-acrosomal regions [18]. In addition, choline acetyltransferase and acetylcholinesterase have been found in ram, rat, and human sperm [19,20]. Furthermore, the effects of ACh and NIC on sperm functions, such as AR, motility, and viability, have been widely reported. However, there are several contradictions between the reports, and hence, more detailed studies to examine the relationship between nAChR and sperm functions are necessary. Although some reports suggest that nAChRs play an important role in mammalian sperm function, which of these subunits is responsible for the function still remains unknown. Many studies have focused on the α7 nAChR subunit (AChRa7) [18,21,22,23] because the neuronal-type of nAChR consists of only a single AChRa7; however, only a few studies have focused on other subunits. Therefore, if the functional role of the other subunits that make up nAChR, and their effects on sperm function via this receptor are clarified, further understanding of the mammalian fertilization and regulation of mammalian sperm function using ACh and NIC may be possible.

From NCBI database (https://www.ncbi.nlm.nih.gov/gene/11448), the mRNA expression level of the ε subunit of nAChR (AChRe) in mouse testes is about 240 times higher than that in the brain, and approximately 10 to 100 times higher than that in other subunits and subtypes. Therefore, in this study, we aimed to (1) determine the presence and level of ACh in female mouse reproductive tissues by measuring ACh concentrations in uterus and oviduct; (2) determine the expression and localization of AChRe in the testes and spermatozoa of mice using western blotting and immunofluorescence; (3) analyze the effects of ACh and NIC on tyrosine phosphorylation, motility, and AR; and (4) determine whether AChRe is responsible for the effects of ACh and NIC on sperm functions through an AChRe-specific antagonist, waglerin-1 (WTX-1).

## 2. Materials and Methods

### 2.1. Ethics Statement

All animal care activities and experiments in this study were conducted in compliance with the Regulations for Animal Experiments and Related Activities at Tohoku University. The study was approved by the Tohoku University Institutional Animal Care and Use Committee (2019-003-02).

### 2.2. Sample Preparation and Measurement of ACh Concentration in Uterus and Oviduct

To measure the concentrations of ACh in uterus and oviduct, we prepared 10 C57BL6/N female mice (aged 6–8 weeks) and half of them were induced to the estrus phase. Estrus phase was induced by superovulation with 5 IU of pregnant mare serum gonadotropin, followed at 48 h interval by 5 IU of human chorionic gonadotropin (hCG) injection. Sixteen hours later, five female mice that had been induced into the estrus phase were sacrificed and five diestrus female mice were also sacrificed at this time, and their uteri and oviducts were collected. Samples were homogenized CHCL_3_/MeOH (2:1, *v*/*v*) and centrifuged at 5100× *g* for 5 min at 4 °C to remove debris. Next, the samples were incubated with dH_2_O for 10 min at room temperature and centrifuged again. The lower (chloroform) organic phase was collected, while the upper phase was re-extracted with the solvent mixture CHCL_3_/MeOH/water (86:14:1, *v*/*v*/*v*). After extraction, the organic phases were combined and dried in a vacuum centrifuge and further dissolved in CHCl_3_/MeOH/water (60:30:4.5, *v*/*v*/*v*) for storing the samples. Measurement of ACh concentrations was performed using a commercially available kit (Acetylcholine Assay Kit; Fluorometric, Cell Biolabs, San Diego, CA, USA) in accordance with the manufacturer’s protocol.

### 2.3. Mouse Sperm Preparation and Capacitation

C57BL6/N male mice were purchased from Japan SLC, Inc. (Shizuoka, Japan). At least three sexually mature mice (aged > 12 weeks) were euthanized, following which the pair of cauda epididymis were collected. Each epididymis was dissected and punctured with a needle. The exposed spermatozoa were placed in 500 µL of human tubal fluid (HTF) medium [24], consisting of 101.6 mM NaCl, 4.7 mM KCl, 0.37 mM K_2_PO_4_, 0.2 mM MgSO_4_·7H_2_O, 2 mM CaCl_2_, 2 mM NaHCO_3_, 2.78 mM glucose, 0.33 mM pyruvate, 21.4 mM sodium lactate, 286 mg/L penicillin G, 228 mg/L streptomycin, and 5 mg/mL fatty acid-free bovine serum albumin. For capacitation, the collected epididymal spermatozoa were incubated for 1 h at 37 °C for under a humidified atmosphere containing 5% CO_2_. After 10 min of diffusion, sperm concentration was adjusted as desired and used for the subsequent experiments. All the experiments were performed using pooled semen samples from at least three mice.

### 2.4. Western Blot Analysis

For analysis of total sperm lysate, spermatozoa suspended in HTF medium were obtained by centrifugation at 7900× *g* for 5 min at 4 °C and washed with phosphate-buffered saline (PBS). The final pellet was resuspended in RIPA buffer (50 mM Tris-HCL, pH 7.6, 150 mM NaCl, 1% Nonidet P-40, 0.5% sodium deoxycholate, and 1% protease inhibitor) (Nacalai Tesque, Kyoto, Japan) and sonicated to extract the proteins. The samples were centrifuged at 7900× *g* for 5 min at 4 °C and the supernatants were collected. Protein concentrations of the supernatants were determined using a bicinchoninic acid protein assay kit (Thermo Fisher Scientific, Waltham, MA, USA). Further, an equal volume of 2× sample buffer (Nacalai Tesque) was added to the sample and the mixture was boiled at 100 °C for 5 min. Proteins were separated by 10% SDS-PAGE (150 V, 0.4 A, 65 min) and transferred to polyvinylidene difluoride membranes. Protein Ladder One Triple-color for SDS-PAGE (Nacalai Tesque) was used as a protein marker. The membranes were blocked for 1 h with Blocking One (Nacalai Tesque) at room temperature and incubated overnight at 4 °C with a 1:1000 dilution of the primary antibody (anti-AChRe) (sc376747, Santa Cruz Biotechnology, Dallas, TX, USA) diluted in Blocking One and PBS mixture (Blocking One:PBS, 1:4). The membranes were washed thrice with PBS-T (PBS containing 0.1% Tween 20) and incubated with a 1:2000 dilution of horseradish peroxidase (HRP)-conjugated anti-mouse IgM antibody (Santa Cruz Biotechnology) for 2 h at room temperature. After two washes, the membranes were treated with Chemilumi One (Nacalai Tesque), and images were obtained using an LAS-3000-mini Lumino Image Analyzer (Fujifilm, Tokyo, Japan).

### 2.5. Immunocytochemistry and Immunohistochemistry

Immunocytochemical analysis was performed to investigate the localization of AChRe in mouse sperm. Epididymal spermatozoa were suspended in HTF medium and collected by centrifugation at 300× *g* for 5 min at 24 °C. Collected sperm cells were fixed with 2% paraformaldehyde (PFA) and permeabilized with 0.2% Triton X-100 in PBS for 15 min at room temperature, followed by centrifugation at 7900× *g* for 5 min at 4 °C. The pellet was washed twice with PBS, blocked with Blocking One for 1 h at room temperature, and incubated overnight at 4 °C with a 1:100 dilution of the primary antibody (anti-AChRe). After three washes with PBS, the suspensions were incubated for 2 h at 4 °C with anti-mouse IgG-Alexa Fluor 555 (Thermo Fisher Scientific), fluorescein isothiocyanate-conjugated peanut agglutinin (FITC-PNA) (J Oil Mills, Tokyo, Japan), and Hoechst 33342 (Thermo Fisher Scientific). Finally, the treated samples were washed, suspended in PBS, mounted on glass slides, and covered with a coverslip. Stained cells were visualized using a fluorescence microscope (BZ-X710, Keyence, Osaka, Japan).

Additionally, we performed immunohistochemistry to examine the localization of AChRe in mouse testes. Similar to the above-mentioned sperm immunofluorescence, AChRe was labeled with Alexa Fluor 555, using the Zenon IgG labeling kit of Molecular Probes (Zenon Alexa Fluor 555 Mouse IgG1 Labeling Kit, Thermo Fisher Scientific). Mouse testes were surgically removed, fixed with 4% PFA in PBS, and embedded in paraffin. Cross sections (4 µm) were mounted on glass slides, deparaffinized, and rehydrated. The antigens were retrieved by immersion in HistoVT One (Nacalai Tesque) for 30 min at 90 °C. After washing, the slides were blocked with Blocking One for 30 min at room temperature and incubated with a primary antibody labeled with Zenon mouse IgG1 labeling reagents, for 2 h at room temperature. After three washes each with PBS-T and PBS, the slides were subjected to second fixation, wherein the tissue sections were fixed in 4% PFA in PBS for 15 min at room temperature, washed with PBS, and then counterstained with FITC-PNA, and Hoechst 33342. Finally, the treated slides were washed with PBS, covered with a coverslip, and mounted with an antifade mounting medium. Stained tissues were visualized using a confocal laser scanning fluorescence microscope (LSM700, Zeiss, Oberkochen, Germany).

### 2.6. Sperm Protein Tyrosine Phosphorylation Analysis

To detect sperm protein tyrosine phosphorylation, equivalent volumes of the sperm suspensions (5 × 10^6^ cell/mL) were divided into microtubules. ACh (Nacalai Tesque) or NIC (Nacalai Tesque) was dissolved in PBS, which was used as the control, at a final concentration of 1, 10, 100, and 1000 µM, and added to the samples. The suspensions were cultured for 3 h at 37 °C under a humidified atmosphere containing 5% CO_2_. After incubation, sperm pellets were collected by centrifugation at 7900× *g* for 5 min at 4 °C. The obtained sperm pellets were washed with TBS and centrifugated again at 7900× *g* for 5 min at 4 °C. After discarding the supernatants, RIPA buffer and 2× sample buffer was added to the pellets for the extraction of proteins. The obtained proteins were separated by 10% SDS-PAGE and transferred to polyvinylidene difluoride membranes. After blocking for 1 h with Blocking One, the membranes were treated overnight at 4 °C either with a 1:10,000 dilution of the mouse monoclonal anti-phosphotyrosine 4G10 antibody (#05-321, Merck Millipore, Darmstadt, Germany) or a rabbit monoclonal anti-alpha-tubulin antibody (ab52866, Abcam, Cambridge, UK) as an internal control. After washing thrice with TBS-T, the membrane was treated with HRP-conjugated anti-mouse or anti-rabbit IgG antibodies (1:2000) for 2 h at room temperature. Detection was performed with Chemi-Lumi Super, and images were obtained using an LAS-3000-mini Lumino Image Analyzer (Fujifilm). Densitometric analyses were performed using the Image Gauge v4.22 analysis software (Fujifilm).

### 2.7. Sperm Motility Assay

To evaluate sperm motility, we prepared 12 µm deep four-chamber slides (SC-20-01-04-B, Leja, Niew-Vennep, The Netherlands). After capacitation treatment, ACh or NIC was added to each suspension at final concentrations of 1, 10, 100, and 1000 µM. Samples were incubated for different time periods at 37 °C under a humidified atmosphere containing 5% CO_2_. Sperm motility analysis was performed on 6 µl aliquots from each tube, at 1, 2, and 3 h. At least 100 spermatozoa in five fields of a chamber were divided into motile and non-motile sperm, and both the percentage of motile sperm and sperm motility parameters were evaluated using a computer-assisted sperm analysis (CASA) system (SMAS, Ditect, Tokyo, Japan). Films were taken for 1 sec, at an interval of once per 1/60 s. The sperm motility parameters evaluated were straight line velocity (VSL, µm/s), curvilinear velocity (VCL, µm/s), linearity (LIN = VSL/VCL × 100, %), amplitude of lateral head displacement (ALH, µm), and head-beat-cross frequency (BCF, Hz). The trajectories of spermatozoa were automatically extracted from the movie and overlaid on the last frame using the CASA system.

### 2.8. Assessment of Mouse Sperm AR

Spermatozoa were capacitated in HTF medium for 1 h and divided into equal volumes in microtubules (5 × 10^6^ cell/mL). ACh and NIC stock solutions were added to each equal volume suspensions at final concentrations of 1, 10, 100, and 1000 µM and incubated for 3 h at 37 °C under a humidified atmosphere containing 5% CO_2_. After incubation, the suspensions were smeared onto glass slides and air-dried. Further, the smears were blocked with Blocking One and treated with FITC-PNA (J Oil Mills) to trace the microscopically acrosome-reacted spermatozoa [25]. Cells were incubated with FITC-PNA and Hoechst 33,342 for 30 min in a light-shielded humidity chamber, washed with PBS, and covered with a coverslip. To investigate the effect of AChRe on AR, we pre-incubated the spermatozoa with 10 µM of waglerin-1 (WTX-1) (Smartox Biotechnology, Saint Egreve, France), an AChRe-specific antagonist, 15 min before the addition of ACh or NIC and AR assessment. WTX-1 was dissolved in dimethyl sulfoxide (DMSO) and cultured with sperm suspensions. All the groups contained 0.1% DMSO. The acrosome-reacted spermatozoa were quantified under a LSM700 (Zeiss). Duplicate counting of at least 100 spermatozoa was performed. Spermatozoa with bright green fluorescence at the acrosome region were counted as “acrosome-intact”, while those with no fluorescence over the acrosomal region were counted as “acrosome-reacted”. The rate of AR was evaluated by calculating the number of acrosome-reacted sperm per total counted sperm.

### 2.9. Statistical Analysis

All experiments were repeated at least three times. Data are presented as mean ± standard deviation (SD). All data followed a normal distribution and variances were homogenous. Statistical analyses were carried out using the Student’s *t*-test (Section 3.1), Dunnett’s test (Section 3.3, Section 3.4, Section 3.5), and the Bonferoni/Dunn test (Figure 7A,B). Values of *p* < 0.05 were considered to indicate significant differences (* *p* < 0.05, ** *p* < 0.01).

## 3. Results

### 3.1. ACh Concentrations in Uterus and Oviduct

We found that the ACh concentrations in the uterus of diestrus and estrus phase were approximately 290 µM and 280 µM, respectively (Figure 1A). The ACh concentration of the estrus phase was significantly higher than that of the diestrus phase (150.2 ± 5.2 µM versus 130.3 ± 8.7 µM) in the oviduct (Figure 1B). This result showed that approximately 100–300 µM of ACh is present in all female reproductive tissues.

### 3.2. Expression of AChRe

Western blotting revealed a specific band around 52 kDa in the lanes for mouse sperm and testes (Figure 2). Molecular weight of this band coincides with that of AChRe.

Immunofluorescence experiments showed that AChRe was expressed in spermatids at specific times of spermatogenesis in the testis, spermatids after step 12, and was localized throughout the sperm (Figure 3). In the sperm, AChRe was localized at the acrosomal region of the sperm (Figure 4). Interestingly, only about 50% of murine spermatozoa showed this immunoreactivity (Table 1). Taken together, these results indicate that AChRe is expressed from the process of spermatogenesis in mouse intratesticular spermatozoa and expressed in about half of epididymal spermatozoa.

### 3.3. Effects of ACh and NIC on Sperm Protein Tyrosine Phosphorylation

To investigate whether ACh or NIC is involved in the sperm protein tyrosine phosphorylation, a major indicator of sperm capacitation, we analyzed the amount of tyrosine phosphorylated proteins with western blotting. Western blot analysis showed no significant difference in sperm protein tyrosine phosphorylation for any concentrations of ACh or NIC (1, 10, 100, 1000 µM) (Appendix A). This result suggests that ACh and NIC exert little effect on tyrosine phosphorylation and sperm capacitation in a broader sense.

### 3.4. Effects of ACh and NIC on Sperm Motility

We evaluated the effects of ACh and NIC on sperm motility, which is essential for successful fertilization in mammals. As the sperm movement shifts to hyperactivated motility, the flagellar bend and beat pattern changes asymmetrically and increases in amplitude, which was reflected in the motility parameters with an increase in VCL and ALH, and a decrease in LIN [26]. CASA analysis showed that LIN tended to increase when spermatozoa were incubated with 1 µM ACh or 10 µM NIC for 3 h, compared with that in the control group. However, there were no significant differences in motility rate, VSL, ALH, and BCF between the control group and any other concentration of the ACh or NIC treated groups at any time point (Figure 5). The motility parameters of spermatozoa showing hyperactivated motility after 3 h of incubation were as follows: 163.3 ± 12.6 in VCL (µm/s), 22.8 ± 0.9 in LIN (%), and 5.7 ± 0.3 in ALH (µm). There were significant differences in VCL (110.1 ± 12.4 in 1 µM ACh) and LIN (32.0 ± 3.2 in 1 µM ACh, and 31.4 ± 2.2 in 10 µM NIC), but they were opposite to the general motility parameters that changed with hyperactivated motility. These results indicate that ACh and NIC did not affect the sperm hyperactivated motility.

### 3.5. Effects of ACh and NIC on AR

The acrosome status was determined by calculating the percentages of sperm with no PNA signal. We found that the AR rate of the control group was 39.4 ± 4.6, while the rates of 1, 10, 100, and 1000 µM of ACh addition groups were 19.6 ± 4.6, 20.2 ± 5.5, 16.5 ± 3.4, and 16.7 ± 2.1, respectively. Additionally, the rates of 1, 10, 100, and 1000 µM NIC addition groups were 22.0 ± 3.4, 17.1 ± 3.8, 16.7 ± 5.6, and 20.0 ± 4.7, respectively. These findings suggest that both ACh and NIC significantly decreased the AR rate in murine sperm (Figure 6). When spermatozoa were pre-incubated with WTX-1, an antagonist of AChRe, prior to incubation with ACh or NIC, the AR rate was significantly higher than that of the ACh or NIC sole addition group and similar to that of control group (Figure 7A,B). These results indicated that both ACh and NIC suppress the AR via AChRe in mice.

## 4. Discussion

Recently, a number of neurotransmitters have been reported to be involved in sperm functions. Several studies have reported that the receptors for neurotransmitters, such as dopamine [7], serotonin [8], gamma-aminobutyric acid (GABA) [9], and neurotensin [10,11] are present on mammalian sperm, and the ligands corresponding to their receptors are thought to regulate sperm capacitation and/or AR. In this study, we focused on ACh, one of its receptors, nAChR, and one of its subunits, AChRe. We selected AChRe because it had a significantly higher level of mRNA expression in mouse testis than any other subunits comprising the nAChR (NCBI database, https://www.ncbi.nlm.nih.gov/gene/11448).

ACh is a neurotransmitter involved in sympathetic, parasympathetic, and motor nerves. Additionally, the ACh synthesized/released by non-neuronal cells and tissues (collectively known as non-neuronal ACh) has been reported to be responsible for local intercellular signaling [12]. In this study, we examined whether non-neuronal ACh is indeed secreted in the uterus and oviduct. Our results indicate the possibility that spermatozoa are exposed to ACh after ejaculation into female reproductive tissues. In addition, choline acetyltransferase and acetylcholinesterase, the enzymes responsible for ACh synthesis and degradation, respectively, have been found in mammalian spermatozoa such as in rams, rats, and humans [19,20]. This suggests that spermatozoa themselves may also get exposed to some ACh-induced action while synthesizing/degrading ACh.

Further, we revealed the expression and localization of AChRe on mouse sperm and testes (Figure 2, Figure 3 and Figure 4). A previous study has reported localization of the nAChR subunit in the midpiece, neck, and post-acrosomal regions of human sperm [18]. However, we revealed that AChRe is localized in the acrosome on mouse sperm. Therefore, we consider that AChRe is closely related to AR in mice, which is essential for successful fertilization. Based on our observations that approximately half of the spermatozoa possessed AChRe and only spermatids after step 12 expressed AChRe in the testes, we consider that AChRe may have a common characteristic in male reproductive tissues and muscle tissues. AChRe is present in skeletal muscle-type nAChRs at the mammalian neuromuscular junction. There are two types of muscle-type nAChRs, which are immature and mature. Immature nAChRs are composed of (α1)_2_ β1 δ γ subunits and are expressed in fetal muscles and atrophic muscles caused by immobilization, burns, or denervation. On the other hand, mature nAChRs consist of (α1)_2_ β1 δ ε subunits with the γ subunit replaced by AChRe [27,28]. Based on these findings, we suggested that the spermatozoa possessing AChRe would be functionally mature, as well as muscle tissue which include AChRe, and may be a high quality of sperm. Interestingly, this hypothesis was supported by the finding that AChRe in intratesticular spermatozoa was only expressed in maturing spermatids from step 12 at the end of spermatogenesis. In addition, almost all of the spermatids in the testis possessed AChRe, while about half of the cauda epididymal spermatozoa possessed AChRe. This may be because spermatozoa undergo some modification during the maturation of spermatozoa in epididymis, and the conversion of AChRe to AChRg occurs in epididymal spermatozoa as it occurs in atrophic muscles, or the removal of the AChRe protein occurs, as many proteins in epididymal spermatozoa are changed during epididymal transit. However, further studies are needed to detect of the differences between sperm with AChRe and without AChRe. This may lead to further understanding of the functional role of nAChRs in fertilization and establishing a new sperm selection method.

Finally, we analyzed the effects of ACh and NIC on sperm functional changes, tyrosine phosphorylation of sperm proteins, sperm motility, and AR, which are essential for mammalian fertilization. It has been reported that various sperm proteins undergo tyrosine phosphorylation, regulated by bicarbonate ions and Ca^2+^ that produce cAMP during the process of sperm capacitation [5,29]. Tyrosine phosphorylation is widely used as a biochemical marker of sperm capacitation. Additionally, capacitated spermatozoa can exhibit a hyperactivated motility with Ca^2+^ influx into the sperm tail, which is necessary to propel them through the highly viscous oviductal fluid and penetrate the granulosa cells and zona pellucida [30,31,32]. When the sperm movement shifts to hyperactivated motility, the flagellar bend and beat pattern changes asymmetrically and acquires a higher amplitude [4], which is reflected in the motility parameters by an increase in VCL and ALH and a decrease in LIN [26]. In this study, we observed that sperm protein tyrosine phosphorylation was affected by neither ACh nor NIC (Appendix A). Moreover, among the sperm motility parameters, there were no significant differences in ALH, and a decrease in VCL and an increase in LIN were observed which is opposite to the change in motility parameters associated with hyperactivated motility. (Figure 5). These results suggest that ACh and NIC do not trigger capacitation or hyperactivation in mice. After capacitation, mammalian spermatozoa undergo a highly modified acrosomal exocytosis phenomenon known as the AR, which is necessary for sperm-oocyte membrane fusion. Our results showed that the addition of at least 1 µM of ACh or NIC to spermatozoa suppresses the AR (Figure 6). When spermatozoa were pre-incubated with WTX-1, an AChRe-specific antagonist, prior to incubation of ACh or NIC, AR rate was not significantly decreased and was similar to that of the control group (Figure 7A,B). These results indicate that ACh and NIC-mediated nAChR signaling suppresses the AR via the AChRe. We found that approximately 100–300 µM ACh was present in the female reproductive tract (Figure 1). However, spermatozoa were affected by ACh from concentrations of at least 1 µM, suppressing the AR. ACh is known to be used in various organs in vivo and to play various roles. In the uterus, ACh has been reported to be used for contraction of uterine muscle, thus intrauterine ACh is used for more than just sperm. Therefore, the ACh concentration in the uterus may not always be uniform at 100–300 µM, and the fact that spermatozoa were affected at ACh concentrations of at least 1 µM could be because sperm may have a high sensitivity to ACh to ensure that they can receive the signal at any environmental concentration in the uterus.

The influx of Ca^2+^ into the spermatozoa is known to be required for inducing AR [4,33]. Briefly, AR may be induced by the elevation of intracellular Ca^2+^ concentration of sperm [34]. nAChRs are representative receptors of the ligand-gated ion channel superfamily and are involved in Na^+^ and Ca^2+^ influx and K^+^ efflux. Thus, nAChR-mediated effects of ACh and NIC may have influenced sperm AR by regulating the intracellular Ca^2+^ concentration. However, in addition to reports that this ligand-gated ion channel is involved in the influx and efflux of Ca^2+^ and other cations, there are numerous reports on the functions of non-neuronal nAChRs in mammals. Therefore, it is possible that nAChRs rather than ion channels may have contributed to our findings. In contrast, tyrosine phosphorylation is regulated by the cAMP or PKA-mediated pathways [35], and ACh and NIC are unlikely to be involved in this pathway. Previous studies have reported elevation of intracellular Ca^2+^ concentrations in human sperm with addition of ACh, and induction of AR in human and mouse spermatozoa by ACh and NIC [22,36,37]. With respect to sperm motility parameters, an increase in human sperm motility by ACh [38] has been reported earlier. On the other hand, although NIC increased the sperm motility in mice [39], this agonist decreased the motility in humans [40]. However, these reports are inconsistent with our results, wherein we observed that the effect of both ACh and NIC on sperm motility was small. The cause of this contradiction is obscure, but we consider that variations owing to inter-species differences, dissimilarity in mouse strains and the analyzed concentrations of ACh and NIC may have led to this. Further research is needed to clarify whether the suppressive effects of ACh and NIC on AR revealed in this study are species-specific, or a common phenomenon that is conserved in other animal species.

Our results showed that ACh and NIC affected the AR in murine sperm via AChRe, but not sperm capacitation or hyperactivated motility. Hence, we consider that while ejaculated spermatozoa are capacitated during their migration through the female reproductive organs, sperm can be influenced by ACh via AChRe to regulate the timing of fertilization such that the AR does not occur before it reaches the oocyte. A recent study reported that during in vivo fertilization in mice, spermatozoa that have induced AR could not pass through the uterotubal junction [41]. This suggests that spermatozoa that have induced AR in the uterus cannot reach the site of fertilization. Thus, intrauterine ACh may play a role in suppressing AR. In addition, in vivo live imaging analysis in mice demonstrated that both spermatozoa with and without acrosome were present in the isthmus of fallopian tube, whereas most spermatozoa in the ampulla did not have acrosome [42]. Furthermore, it has been reported that most mouse spermatozoa fertilizing in vivo are likely to induce AR before attaching to zona pellucida [43]. These findings suggest that the AR is induced during the migration from isthmus to ampulla of the oviduct, indicating that the timing of sperm AR in the oviduct is elaborately regulated. Several factors in the oviduct, such as neurotensin and GABA, are already known to promote AR [9,11]. Thus, ACh may work together with other enhancers of AR to regulate sperm-inducing AR at the optimal time for fertilization. Indeed, a previous report suggested that progesterone-induced capacitation and hyperactivated motility were competitively inhibited by GABA through the GABA_A_ receptors, ligand-gated ion channel receptors, similar to nAChRs, in hamsters [44]. Overall, the sperm functions in the process of mammalian fertilization may be complexly modulated by a variety of factors, including ACh.

## 5. Conclusions

In summary, we focused on the AChRe among the subunits of nAChRs and revealed the expression and localization of AChRe in the testes and spermatozoa of mice. In addition, we confirmed that both ACh and NIC suppress sperm AR via AChRe. Furthermore, we measured the ACh concentration of the female reproductive tissues. Our findings suggest that mouse spermatozoa are exposed to 100–300 µM of ACh during their migration from the uterus to oviduct, and ACh may regulate the timing of fertilization to avoid inducing AR via AChRe. Further understanding of the molecular mechanisms that regulate sperm functions via ACh-AChRe signaling can provide new insights into sperm physiology and the process of fertilization in mammals. Additionally, it may be possible to control sperm function using ACh, NIC, or WTX-1 (AChRe antagonist), and to select high quality sperm by the presence or absence of AChRe. This may aid in the establishment of a new method of superior sperm selection, infertility treatment, and improvement of fertilization rate in mammals.

## Figures and Tables

**Figure 1 biology-10-00046-f001:**
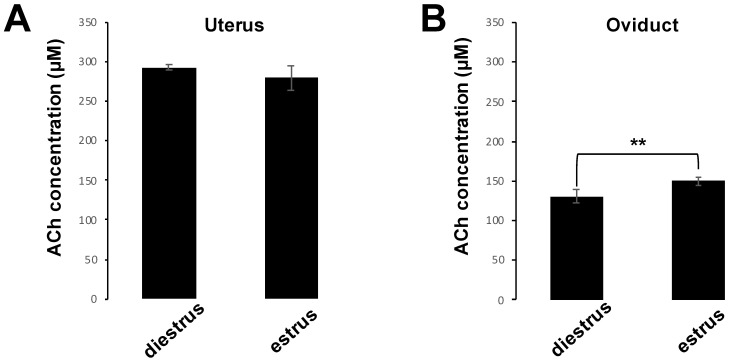
ACh concentrations in mouse uterus and oviduct during diestrus and estrus phase. (**A**,**B**) Acetylcholine (Ach) concentration (µM) in uterus (**A**), and oviduct (**B**). Data shown as mean ± *SD* (*n* = 5; ** *p* < 0.01).

**Figure 2 biology-10-00046-f002:**
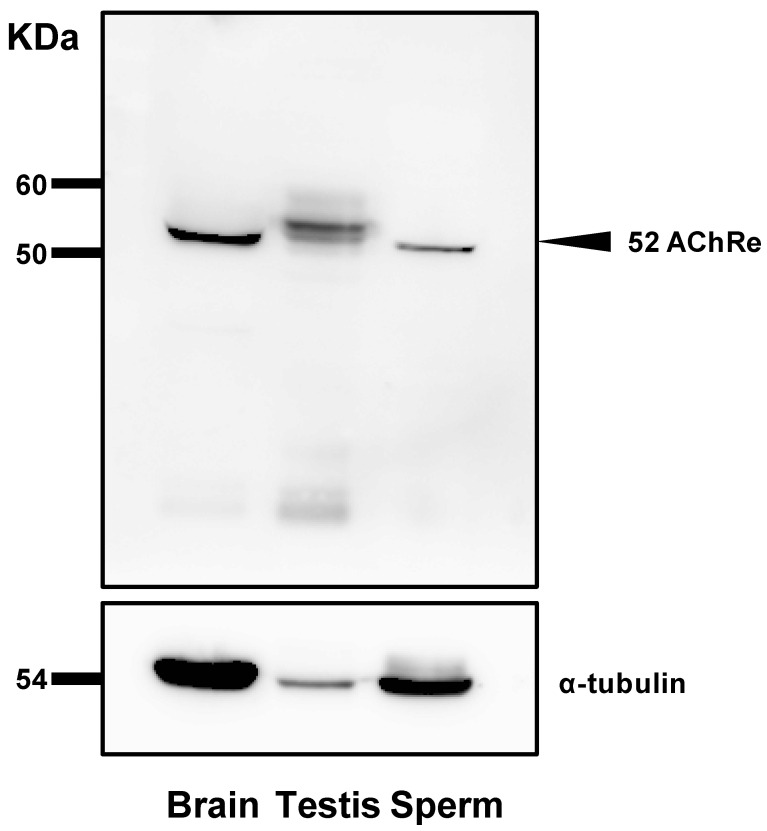
Acetylcholine receptor ε subunit (AChRe) expression analysis by western blotting. Protein extracts from mouse sperm, testes, and positive control (brain) samples (30 µg) were separated by 10% SDS-PAGE followed by detection with specific antibodies. Note the specific band corresponding to AChRe at the predicted size.

**Figure 3 biology-10-00046-f003:**
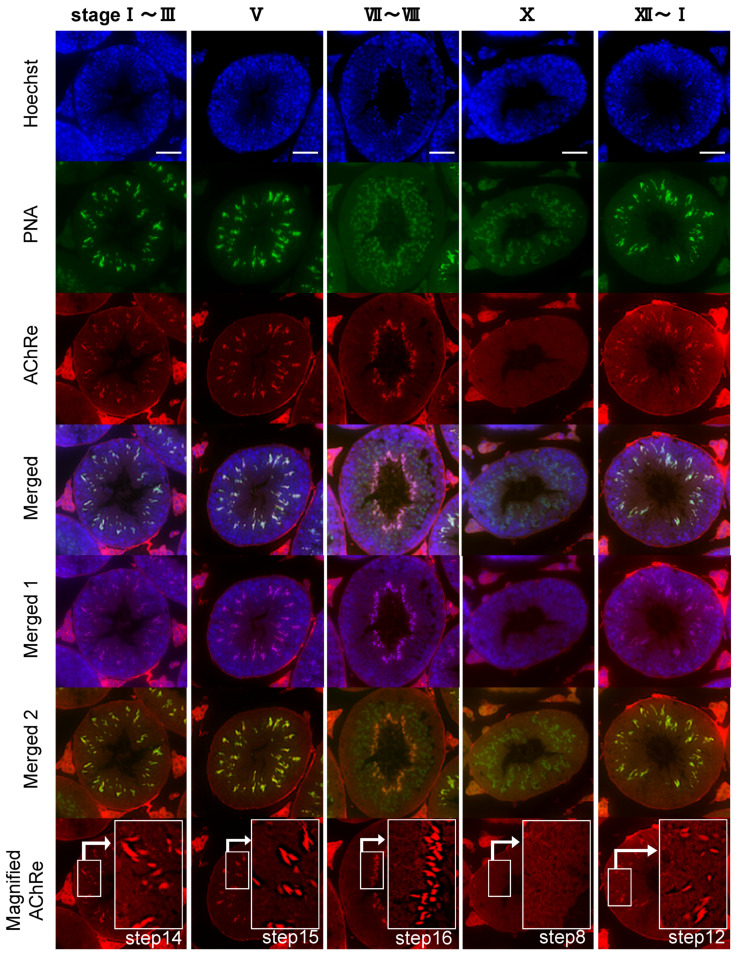
Testis AChRe localization analysis by immunocytochemistry. (Merged 1): Merged image of nuclear and AChRe. (Merged 2): Merged image of acrosome and AChRe. Note the immunoreactivity detected in spermatids from step 12 onwards. AChRe was expressed in spermatids from the end of spermatogenesis process to just before ejaculation. Blue, nuclear (Hoechst 33342); green, acrosome (FITC-PNA); red, AChRe; bars = 50 µm.

**Figure 4 biology-10-00046-f004:**
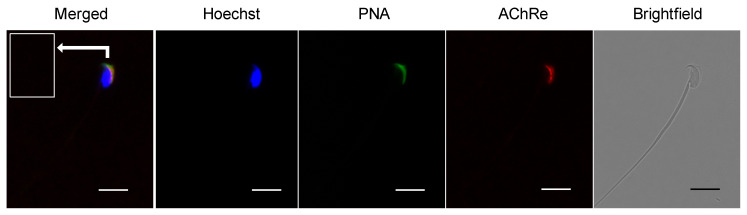
Mouse sperm AChRe localization analysis by immunocytochemistry. Note the immunoreactivity, which was detected at the anterior acrosomal region overlapping with PNA lectin signal. Blue, nuclear (Hoechst 33342); green, acrosome (FTIC-PNA); red, AChRe; bars = 10 µm.

**Figure 5 biology-10-00046-f005:**
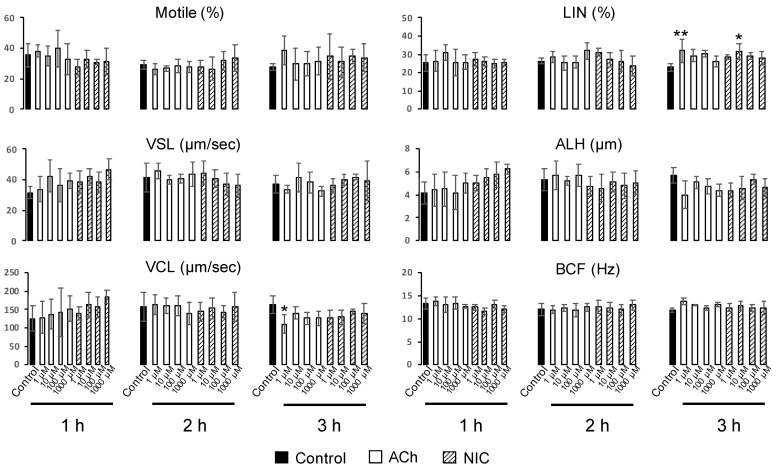
Computer-assisted sperm analysis (CASA) measurements of the effects of ACh and NIC on mouse sperm motility. Data shown as mean ± *SD* (*n* = 4; * *p* < 0.05; ** *p* < 0.01). VSL, straight-line velocity; VCL, curvilinear velocity; LIN, linearity; ALH, amplitude of lateral head displacement; BCF, beat cross frequency.

**Figure 6 biology-10-00046-f006:**
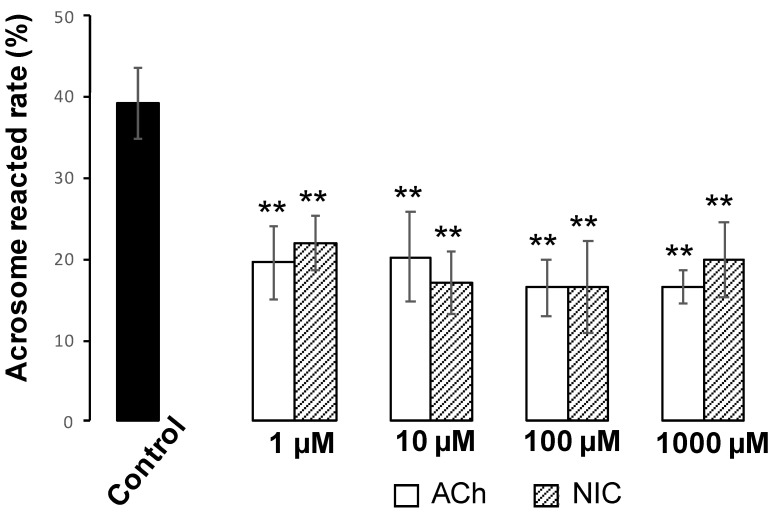
Effects of ACh and nicotine (NIC) on the sperm acrosome reaction. ACh and NIC administration suppresses AR. Mouse spermatozoa were capacitated in human tubal fluid (HTF) medium for 1 h. Phosphate-buffered saline (PBS) or various concentrations of ACh or NIC were added to equivalent aliquots of the suspension (200 µL, 5 × 10^6^ cell/mL) and cultured for 3 h. All the groups contained 0.1% dimethyl sulfoxide (DMSO). Sperm acrosomal disappearance rates were evaluated by calculating the number of PNA-negative sperm out of the total counted sperm. Data shown as mean ± *SD* (*n* = 3; ** *p* < 0.01).

**Figure 7 biology-10-00046-f007:**
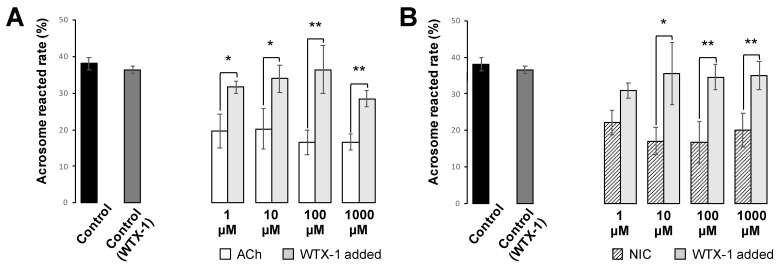
Effect of AChRe antagonist on sperm AR suppression caused by ACh and NIC. (**A**,**B**) Administration of WTX-1, an AChRe-specific antagonist, 15 min before the addition of ACh and NIC, inhibited the effects of ACh (**A**) and NIC (**B**) on AR. All the groups contained 0.1% DMSO. The experiments in Figure 6 and Figure 7 were performed on the same day, and the data for control, ACh, and NIC treated group are the same as in Figure 6. Sperm acrosomal disappearance rates were evaluated by calculating the number of PNA-negative sperm out of the total sperm. Data shown as mean ± *SD* (*n* = 3; * *p* < 0.05; ** *p* < 0.01).

**Table 1 biology-10-00046-t001:** AChRe expression rate in the mouse sperm. Each group was made up of a sperm mixture sample from three male mice. These three different sperm samples were prepared and tested in total and the average of the three trials was calculated.

	#1	#2	#3	AVE	
AChRe retain sperm	45	58	60		
Total sperm	96	103	101		
AChRe expression rate (%)	46.9	56.3	59.4	54.2	

## Data Availability

Data is contained within the article.

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
