# Peer review of "Expression and Possible Role of Nicotinic Acetylcholine Receptor ε Subunit (AChRe) in Mouse Sperm"

_biology, 2021, doi:10.3390/biology10010046_

Round 1

Reviewer 1 Report

Makino et al. focused on analyzing the role of the nicotinic acetylcholine receptor ε subunit (nAChRe) in mouse sperm using ACh, nicotine (as the receptor agonist), WTX-1 (as the nAChRe-specific antagonist), and anti-nAchRe antibody. The authors found that the concentration of ACh in the uterus is higher than in the oviduct and nAChRe is localized in the sperm acrosome. Moreover, they found that ACh and nicotine inhibited the sperm acrosome reaction. The authors concluded that nAChRe may play an important role in sperm formation and acrosome reaction and that may establish a method to select superior sperm via nAChRe. The authors should pay more attention to the text, as there were many errors in the manuscript (e.g., materials and methods, figures, grammar etc.) and reconsider the structure of the manuscript. Reviewer recommends that the authors use another editing service company to correct the errors. The authors should address the reviewer’s comments as follows and then re-submit this manuscript.

Major points

  1. Figure 2: There is no indication of the data. Please check the uploaded data.
  2. Figures 3 and 4: First, the authors should change the order of figures because it is more straightforward to explain testis data first before sperm data. Second, the authors should indicate AChRe localization of sperm after the acrosome reaction. Third, although all of PNA-positive sperm had AChRe in the testis, about a half of epididymal sperm had AChRe. The authors should discuss the change of localization.  
  3. Figure 5: First, the authors should indicate the whole blotting image of tyrosine phosphorylation of sperm. Second, the authors should be consistent in the order of NIC and ACh on the upper and lower panels. It would be easier for readers to understand the results. Third, please remove p-values from the figure legend. Fourth, these results are negative data so please move them to a supplemental figure. 
  4. Figure 6 and lines 282 and 353-354: There are significant differences in the linearity of sperm motility after 3 hours incubation. However, the authors mentioned that ACh and NIC did not have a significant effect on sperm motility in line 282. This description is not correct. The authors should mention the correct interpretation of the results. 
  5. Figure 7: Please indicate total number of sperm observed. The authors showed that the ACh concentration is 100-300 uM in the female reproductive tract (figure 1). However, figure 7 indicated that the acrosome reaction is inhibited by a lower concentration of ACh (1 uM~). The authors should discuss this difference between in vivo and in vitro.   
  6. Figure 8 and line 294: The authors may use the same data of ACh and NIC additions in figures 7 and 8. If not, the authors should indicate the real percentage of AR rate in line 294. The reviewer recommends to merge figure A with B.
  7. Lines 352-361: The authors should revise the discussion section. Figures 6-8 are especially important for this manuscript. The reviewer requests the authors to mention and explain the following two points: (a) Change of AChRe localization from testicular sperm to epididymal sperm. (b) Effect of ACh on acrosome reaction between in vivo and in vitro
  8. Lines 19 and 33-34: The authors mentioned that AChRe plays an important role in the mammalian sperm formation and AR. However, ACh addition did not inhibit the acrosome reaction and sperm formation in all of the sperm. This means that AChRe is not essential for sperm acrosome reaction and sperm formation. Therefore, this sentence is not correct. The authors should revise the description correctly.
  9. Lines 21 and 413: The authors did not analyze the fertilizing capability of AChRe-positive and -negative sperm. Therefore, the authors do not refer directly to the relationship between AChRe localization and sperm fertilizing competency. The reviewer recommends to perform IVF assay using AChRe-positive and -negative sperm. 

Minor points

  1. The authors should be consistent in the use of words: spermatozoa and sperms, and nAChR and AChR.
  2. The authors should be consistent in the notation of information in the materials and methods section. In general, the order of information is model, company name, city, and country. Please check the journal policy.
  3. Lines 67-69: ‘’many mammalian spermatozoa’’ is not correct. The authors should fix this sentence. 
  4. Lines 81 and 314: Please indicate the link of NCBI database.
  5. Line 134: The anti-AChRe antibody (sc-376747) can be purchased from Santa Cruz Biotechnology. 
  6. Lines 230-231: Please indicate real values including SD. 
  7. Line 232: ‘’all’’ overstates the result because the authors do not analyze the ACh concentration in the ovary.
  8. Line 404: ‘’the first time’’ should be removed from the text. Because a research article is described as only a novelty of research.
  9. Figure 4 legend: ‘’expresed’’ is not correct.

Author Response

Response to Reviewer 1 Comments

To Reviewer 1

Thank you for providing constructive and valuable comments regarding the improvement of our original manuscript. We have carefully reviewed comments and have revised the manuscript accordingly. We hope that our revised version is suitable for publication and look forward to hearing from you. Our responses to your comments are as follows:

Major points

  1. Figure 2: There is no indication of the data. Please check the uploaded data.

>> It seems that the PDF version did not upload properly. Please see the re-uploaded version.

  1. Figures 3 and 4: First, the authors should change the order of figures because it is more straightforward to explain testis data first before sperm data. Second, the authors should indicate AChRe localization of sperm after the acrosome reaction. Third, although all of PNA-positive sperm had AChRe in the testis, about a half of epididymal sperm had AChRe. The authors should discuss the change of localization.

>> First; We have changed the order of the figures. Please see L283-297.

>> Second; To clarify where AChRe is localized on the sperm, we stained the acrosome using FITC-PNA. It was classified into the following four types; both PNA-AChRe positive, PNA-positive but AChRe negative, PNA-negative but AChRe positive and neither PNA-AChRe negative. Thank you for your suggestion, and we are sure that we have to incorporate your valuable feedback in our future work.

>> Third; We have added a discussion on the changes in AChRe expression. Please see L416-421 .

  1. Figure 5: First, the authors should indicate the whole blotting image of tyrosine phosphorylation of sperm. Second, the authors should be consistent in the order of NIC and ACh on the upper and lower panels. It would be easier for readers to understand the results. Third, please remove p-values from the figure legend. Fourth, these results are negative data so please move them to a supplemental figure.

>> Accordingly, we have revised all your points from the first to the third and moved them to the supplemental materials.

  1. Figure 6 and lines 282 and 353-354: There are significant differences in the linearity of sperm motility after 3 hours incubation. However, the authors mentioned that ACh and NIC did not have a significant effect on sperm motility in line 282. This description is not correct. The authors should mention the correct interpretation of the results.

>> Thank you for pointing that out. We agree that it is not correct to say ACh and NIC did not have a significant effect on sperm motility. We have revised the text about the sperm motility. Please see L332-335 and 436-439.

  1. Figure 7: Please indicate total number of sperm observed. The authors showed that the ACh concentration is 100-300 uM in the female reproductive tract (figure 1). However, figure 7 indicated that the acrosome reaction is inhibited by a lower concentration of ACh (1 uM~). The authors should discuss this difference between in vivo and in vitro.

>> In materials and methods of assessment of mouse sperm AR, we stated that we counted at least 100 sperms.

>> We have added a discussion on the differences between in vivo and in vitro. Please see L457-465.

  1. Figure 8 and line 294: The authors may use the same data of ACh and NIC additions in figures 7 and 8. If not, the authors should indicate the real percentage of AR rate in line 294. The reviewer recommends to merge figure A with B.

>> As you pointed out, the data in figure 7 and 8 are the same. Thank you for the recommendation. However, if we merge Figure 8A and B, it will complicate the marking of significant differences and graphing, so please leave it as it is.

  1. Lines 352-361: The authors should revise the discussion section. Figures 6-8 are especially important for this manuscript. The reviewer requests the authors to mention and explain the following two points: (a) Change of AChRe localization from testicular sperm to epididymal sperm. (b) Effect of ACh on acrosome reaction between in vivo and in vitro.

>> Thank you for your important suggestions. As mentioned above, discussion about (a) Change of AChRe localization from testicular sperm to epididymal sperm was added to L416-421, and (b) Effect of ACh on acrosome reaction between in vivo and in vitro was added to L457-465.

  1. Lines 19 and 33-34: The authors mentioned that AChRe plays an important role in the mammalian sperm formation and AR. However, ACh addition did not inhibit the acrosome reaction and sperm formation in all of the sperm. This means that AChRe is not essential for sperm acrosome reaction and sperm formation. Therefore, this sentence is not correct. The authors should revise the description correctly.

>> We have revised AChRe, weakened expressions. Please see L19 and 34.

  1. Lines 21 and 413: The authors did not analyze the fertilizing capability of AChRe-positive and -negative sperm. Therefore, the authors do not refer directly to the relationship between AChRe localization and sperm fertilizing competency. The reviewer recommends to perform IVF assay using AChRe-positive and -negative sperm.

>> Thank you for your important suggestions. We also thought we needed to do some IVF experiments. However, we are in the process of investigating whether it is possible to sort sperm according to AChRe-positive and -negative, and this will be an issue for the future research.

Minor points

  1. The authors should be consistent in the use of words: spermatozoa and sperms, and nAChR and AChR.

>> we have unified our use of words: spermatozoa, and nAChR.

  1. The authors should be consistent in the notation of information in the materials and methods section. In general, the order of information is model, company name, city, and country. Please check the journal policy.

>> We have fixed it.

  1. Lines 67-69: ‘’many mammalian spermatozoa’’ is not correct. The authors should fix this sentence.

>> We have changed this sentence to “some mammalian spermatozoa”. Please see L77.

  1. Lines 81 and 314: Please indicate the link of NCBI database.

>> Accordingly, we have added the link of NCBI datebase. Please see L91 and 389.

  1. Line 134: The anti-AChRe antibody (sc-376747) can be purchased from Santa Cruz Biotechnology.

>> Thank you for pointing this out. I made a mistake. We have fixed it.

  1. Lines 230-231: Please indicate real values including SD.

>> We have indicated real values including SD. Please see L256.

  1. Line 232: ‘’all’’ overstates the result because the authors do not analyze the ACh concentration in the ovary.

>> As indicated, we have removed the word “all”. Please see L257.

  1. Line 404: ‘’the first time’’ should be removed from the text. Because a research article is described as only a novelty of research.

>> Thank you for the advice. We have removed it. Please see L514.

  1. Figure 4 legend: ‘’expresed’’ is not correct.

>> We have fixed the word. Thank you for your pointing.

Reviewer 2 Report

I have reviewed the manuscript titled “Expression and Possible Role of Nicotinic Acetylcholine Receptor ε Subunit (AChRe) in Mouse Sperm”. Despite the topic being very interesting, several gaps were present in the experimental design and in the description of materials and methods.

Lines 97-98 How many females were included in the study? How many females were included in each group?

Lines 97-101 When were the females sacrificed in diestrus phase?

Lines 112-113 How many males were included in the study? How many males were included in each group?

Line 130 “the mixture was boiled for 5 min”. It would be good to indicate the precise temperature.

Lines 131-132 It would be appropriate to indicate the conditions of the electrophoretic run (volts, amper time, etc.)

Line 134 With the guidance provided by the authors (anti-AChRe; sc376747, Abcam, Cambridge, US), I did not find the anti-AChRe antibody that reacts with mouse. Please, provide more details regarding the antibody anti-AChRe used.

Lines 123-140 Please, provide information regarding the marker used in western blot analysis.

Line 212 Please, provide dosage information for waglerin-1 (WTX-1)

Lines 222-226 Only parametric tests were used in the statistical analysis however normal distribution was not assessed. It would be appropriate to evaluate the normal distribution and use the appropriate statistical test.

In figure 3 It would be good to report the letters in each image to facilitate the understanding of the figure.

In table 1 what groups are reported (#1, #2, #3)? How many mice belong to each group?

In figures 5, 7 and 8, n = 4 and n = 3 refers to the number of mice included in each group?

in my opinion, the entire manuscript requires a major re-evaluation, in particular the authors should put more attention and details in the description of materials and methods and in the presentation of  the results.

Author Response

Response to Reviewer 2 Comments

To Reviewer 2

We sincerely appreciate your careful reading of our manuscript and valuable comments. We have carefully reviewed comments and have revised the manuscript accordingly. We hope that our revised version is suitable for publication and look forward to hearing from you. Our responses to your comments are as follows:

  1. Lines 97-98 How many females were included in the study? How many females were included in each group?

>> In this study, each group included 5 female mice, for a total of 10 female mice in one experimental trial (n=1). We have added detailed information to the text. Please see L97-98 and L101-102.

  1. Lines 97-101 When were the females sacrificed in diestrus phase?

>> 16 h after the injection of hCG, diestrus female mice were sacrificed at the same time as the female mice that induced estrus phase. We have described it in more detail in the text. Please see L101-102.

  1. 112-113 How many males were included in the study? How many males were included in each group?

>> In this study, we collected epididymal caudal sperm from at least three male mice and mixed them for our experiments. We have added detailed information to the text. Please see L113.

  1. Line 130 “the mixture was boiled for 5 min”. It would be good to indicate the precise temperature.

>> Thank you for pointing that out. Accordingly, we have added the precise temperature. Please see L132.

  1. Lines 131-132 It would be appropriate to indicate the conditions of the electrophoretic run (volts, amper time, etc.)

>> As you indicated, we have added the conditions of the electrophoretic run. Please see L133.

  1. Line 134 With the guidance provided by the authors (anti-AChRe; sc376747, Abcam, Cambridge, US), I did not find the anti-AChRe antibody that reacts with mouse. Please, provide more details regarding the antibody anti-AChRe used.

>> >> Thank you for pointing this out. I made a mistake. It was purchased from Santa Cruz Biotechnology. We have fixed it.

  1. Lines 123-140 Please, provide information regarding the marker used in western blot analysis.

>> As indicated, we added the information of protein marker. Please see L134-135.

  1. Line 212 Please, provide dosage information for waglerin-1 (WTX-1)

>> Thank you for your suggestion. We have added the information of actual dosage used in this study.

  1. Lines 222-226 Only parametric tests were used in the statistical analysis however normal distribution was not assessed. It would be appropriate to evaluate the normal distribution and use the appropriate statistical test.

>> All date had been checked for normal distribution and confirmed to follow a normal distribution. Due to the small number of date (n=3, 4, and 5), we also conducted nonparametric tests, and we have obtained the same results.

  1. In figure 3 It would be good to report the letters in each image to facilitate the understanding of the figure.

>> Thank you for the advice. We have fixed the image. Please see L256-257.

  1. In table 1 what groups are reported (#1, #2, #3)? How many mice belong to each group?

>> Each group indicates that they are experimenting with a mixture of sperm form three male mice. We are experimenting with sperm mixtures from three male mice of different individuals in #1, #2, and #3 respectively. We have revised the text. Please see L261-262.

  1. In figures 5, 7 and 8, n = 4 and n = 3 refers to the number of mice included in each group?

>> No, it’s not. A single experimental trial using sperm mixtures from at least three male mice is represented as n=1. The n=3 and n=4 refer to the number of trials.

Round 2

Reviewer 1 Report

The reviewer confirmed the authors’ responses in the revised manuscript. However, the reviewer found minor points as follows. The authors should address the reviewer’s comments. Please reconfirm these companies' information.

  1. Line 129: This should be ‘’Nacalai Tesque, Kyoto, Japan’’.
  2. Line 143: This should be ‘’Fujifilm, Tokyo, Japan’’.
  3. Line 154: This should be ‘’J Oil Mills, Tokyo, Japan’’.
  4. Line 157: This should be ‘’BZ-X710, Keyence, Osaka, Japan’’.
  5. Lines 185-186: This should be ‘’ab52866, Abcam, Cambridge, UK’’.
  6. Lines 193-194: This should be ‘’SC-20-01-04-B, Leja, Nieuw-Vennep, Netherlands’’.
  7. Lines 199-200: This should be ‘’SMAS, Ditect, Tokyo, Japan’’.
  8. Line 219: This should be ‘’LSM700, Zeiss, Oberkochen, Germany’’.
  9. Figure 2: The authors should quote this in the text. The reviewer could not find this sentence and the data in the revised manuscript. This data can be found in the non-published file. 
  10. Lines 279-281: The authors should indicate the motility parameters of hyperactivated spermatozoa. It is easy for readers to understand the interpretation of figure 5.
  11. Figure 7: The authors should indicate to use the same date of revised figure 6 in the figure legend. 

Author Response

Response to Reviewer 1 Comments (2nd-round review)

To Reviewer 1

First and foremost, thank you for taking the time out of your schedule to do the peer review. And I apologize for submitting the manuscript directly to you, as I did not fully understand how to do when I replied the first time.

Thank you for your minor point suggestions. We have revised the manuscript accordingly. We hope that our revised version is suitable for publication and look forward to hearing from you. Our responses to your comments are as follows:

  1. Line 129: This should be ‘’Nacalai Tesque, Kyoto, Japan’’.
  2. Line 143: This should be ‘’Fujifilm, Tokyo, Japan’’.
  3. Line 154: This should be ‘’J Oil Mills, Tokyo, Japan’’.
  4. Line 157: This should be ‘’BZ-X710, Keyence, Osaka, Japan’’.
  5. Lines 185-186: This should be ‘’ab52866, Abcam, Cambridge, UK’’.
  6. Lines 193-194: This should be ‘’SC-20-01-04-B, Leja, Nieuw-Vennep, Netherlands’’.
  7. Lines 199-200: This should be ‘’SMAS, Ditect, Tokyo, Japan’’.
  8. Line 219: This should be ‘’LSM700, Zeiss, Oberkochen, Germany’’.

>> We have revised these points. Thank you for pointing this out and describing it in detail.

  1. Figure 2: The authors should quote this in the text. The reviewer could not find this sentence and the data in the revised manuscript. This data can be found in the non-published file.

>> It seems that the problem occurs when the editor converts the submitted word file to PDF. In this revised manuscript, I reacquired the image data and added the new one on the manuscript. If the problem occurs again with this revised manuscript, I will contact the editor. Please check L279-280 again.

  1. Lines 279-281: The authors should indicate the motility parameters of hyperactivated spermatozoa. It is easy for readers to understand the interpretation of figure 5.

>> As you indicate, we have added the numerical parameter data for VCL, LIN, and ALH of hyperactivated spermatozoa.

  1. Figure 7: The authors should indicate to use the same date of revised figure 6 in the figure legend.

>> Thank you for your suggestion. We have added the information you mentioned  to the legend of Figure 7.

Reviewer 2 Report

I would like to thank the authors who answered all my questions completely.
In my opinion, the manuscript can be published in biology.

Author Response

To reviewew 2

Thank you for taking time out of your busy schedule to review our manuscript.

Thanks to your suggestion, I think our original manuscript has been further improved.

We would like to work hard on further research in the future.